# Blind Deblurring of Saturated Images Based on Optimization and Deep Learning for Dynamic Visual Inspection on the Assembly Line

**Bodi Wang [1], Guixiong Liu [1,*] and Junfang Wu [2,3]**

1   School of Mechanical and Automotive Engineering, South China University of Technology, Guangzhou 510640, China; merobbiewong@mail.scut.edu.cn
2   School of Physics, South China University of Technology, Guangzhou 510640, China; wujf@scut.edu.cn
3   Guangdong Key Laboratory of Modern Geometry and Mechanics Metrology Technology, Guangzhou 510405, China
*   Correspondence: megxliu@scut.edu.cn

**Abstract:** Image deblurring can improve visual quality and mitigates motion blur for dynamic visual inspection. We propose a method to deblur saturated images for dynamic visual inspection by applying blur kernel estimation and deconvolution modeling. The blur kernel is estimated in a transform domain, whereas the deconvolution model is decoupled into deblurring and denoising stages via variable splitting. Deblurring predicts the mask specifying saturated pixels, which are then discarded, and denoising is learned via the fast and flexible denoising network (FFDNet) convolutional neural network (CNN) at a wide range of noise levels. Hence, the proposed deconvolution model provides the benefits of both model optimization and deep learning. Experiments demonstrate that the proposed method suitably restores visual quality and outperforms existing approaches with good score improvements.

**Keywords:** visual inspection; image deblurring; blur kernel; deconvolution; deep learning

## 1. Introduction

Visual inspection has been extensively investigated and is a well-established approach for a myriad of engineering applications. In dynamic scenes, visual inspection helps to improve the efficiency of monitoring tasks, but low visual quality may undermine its effectiveness [1–4]. For instance, in manufacturing environments, the object under inspection moves along the assembly line during exposure time, resulting in motion blur. Consequently, latent images are degraded by effects such as smears on sharp edges and loss of semantic information. Hardware-based approaches can be applied to compensate for visual quality. However, such methods are costly and require manual adjustments, and the devices often do not provide setup flexibility [5]. Alternatively, image deblurring can be employed, as it reverses degradation and restores the latent image from a blurred one.

Image deblurring can be either blind or non-blind, where the former type operates with unknown kernel. Therefore, blind image deblurring is equivalent to inferring two functions from their product, being an ill-posed problem in general. Recently, blind image deblurring methods have relied on the image prior-based regularization of objective functions to guide deconvolution and alternately estimate the blur kernel and intermediate latent images [6,7]. Once the blur kernel is obtained, blind image deblurring reduces to its non-blind counterpart. Unlike conventional blind image deblurring, the challenges of deblurring for dynamic visual inspection are threefold:

1.  Dynamic visual inspection inherently poses a blind image deblurring problem that should include blur kernel estimation and deconvolution modeling. Still, linear motion of inspected objects in structured environments (e.g., assembly lines) allows to parameterize the blur kernel.
2.  Conventional model optimization relies on a handcrafted prior, which is often sophisticated and non-convex. Moreover, the tradeoff between computation time and recovered visual quality should be considered, as image deblurring for dynamic visual inspection requires fast optimization.
3.  In the case considered for this study, the sheet metal of objects for dynamic visual inspection might intensely reflect light, resulting in a blurred image with saturated pixels, which affect the goodness of fit in data fidelity. Consequently, deblurring based on linear degradation models might fail due to the appearance of severe ringing artifacts.

We focus on image deblurring to improve the visual quality and mitigate motion blur for dynamic visual inspection. Degradation of dynamic visual inspection is mainly attributed to linear motion blur, whose kernel exhibits a sinc-shaped ripple in the transform domain and hence can be characterized by the blur angle and blur length [8]. Considering a fixed heading of the inspected object with respect to the camera, the blur angle can be considered as a known parameter, and the blur length is estimated according to extracted features in the transform domain. Then, variable splitting can be applied to the deconvolution model for its decoupling into data fidelity and regularization terms. To maximize the restored visual quality, the data fidelity term is associated with the mask specifying inliers and outliers via variational expectation–maximization (VEM) [9], and the regularization term is learned via the fast and flexible denoising convolutional neural network (CNN), or fast and flexible denoising network (FFDNet) [10], at a wide range of noise levels. Different from measures that consider an original clear image as ground truth, we evaluate the deblurring performance from the structural similarity index (SSIM) and feature similarity index (FSIM) between the blurred and re-blurred images, as no ground truth is available in dynamic visual inspection.

The remainder of this paper is organized as follows. Section 2 presents a survey of related work on blur kernel estimation, deconvolution modeling, and outlier handling. The degradation model and blur kernel model for dynamic visual inspection are presented in Section 3. In Section 4, we introduce the proposed blind deblurring method of saturated images for dynamic visual inspection and report its validation both quantitatively and qualitatively in Section 5. Finally, we draw conclusions in Section 6.

## 2. Related Work

As we split blind image deblurring into blur kernel estimation and deconvolution modeling, we performed a review of these two topics. Given the need for high restored visual quality during dynamic visual inspection, we also reviewed methods for handling outliers.

### 2.1. Blur Kernel Estimation

Blind image deblurring depends on blur kernel estimation. As both the blur kernel and intermediate latent image are unknown, an image prior is utilized to constrain the solution space and enable blind image deblurring. The image prior describes the nature of an image and can adopt approaches such as heavy-tailed distribution [5,11] and $L_0$ regularization [12,13] or channel prior [14,15]. These methods model the blur kernel as a projection transform and estimate it in a coarse-to-fine process. This process, however, is not suitable for the linear motion blur appearing during dynamic visual inspection [16]. In fact, linear motion blur in the transform domain is identifiable without an image prior and determined by the blur angle and blur length. Thus, blur kernel estimation in linear motion relies on specific features from the transform domain [8,17].

Dash and Majhi [18] found the blur angle according to the frequency response of a blurred image filtered by a 2D Gabor mask, and a radial basis function network retrieved the relationship between the Fourier coefficients and blur length. Oliveira et al. [16] fit a third-order polynomial to approximate the modified Radon transform and assumed that the blur angle maximizes its mean squared error.

Then, they computed the Radon transform along the motion direction to estimate the blur length. Lu et al. [6] found that a restored image has its most sparse representation at the true blur angle, and in their approach, they captured sinc-shaped components in the Fourier domain to measure the blur length. Yan and Shao [19] classified the blur type on the discriminative feature space of a deep neural network and used a general regression neural network to estimate the blur kernel parameters. Lu et al. [20] mined the linear motion blur kernel features in CNNs and mapped them into the blur angle and blur length using a support vector machine.

To collapse 2D data into one dimension (i.e., to align the motion direction with the horizon), several methods rotate the blurred image, its derivative, or its axis using the estimated blur angle [6,8,18]. However, rotation-induced interpolation causes out-of-range pixels, thus hindering the subsequent blur length estimation [21]. In addition, rotation might lead to error propagation given the dependence between blur length and blur angle during estimation [16,17].

## 2.2. Deconvolution Modeling

Deconvolution relies on the estimated blur kernel for performing non-blind image deblurring. The Richardson–Lucy (RL) [22] and Wiener [23] deconvolutions are simple and effective methods but suffer from oversmoothed edges and ringing artifacts. Model optimization methods rely on image prior that is mainly used to characterize local smoothness (e.g., total variation (TV) norm [24,25], hyper-Laplacian (HL) prior [26,27]) and nonlocal self-similarity (e.g., iterative decoupled deblurring-block matching and 3D filtering (IDD-BM3D) [28], nonlocally centralized sparse representation (NCSR) [29], multi scale-weighted nuclear norm minimization (MS-WNNM) [30]) of images. However, optimization is often time consuming, and the solution can reach a local minimum for image prior-based regularization, which may not be sufficiently strong [31].

Recently, deep learning has been applied to low-level vision tasks [32–34]. Xu et al. [35] used the singular value decomposition of a 2D pseudoinverse kernel to initialize the weights of a network, which should be retrained for different blur kernels. To handle different blur kernels in a unified framework, Ren et al. [36] used a neural network motivated by the generalized low-rank approximation of a set of blur kernels, and similarly exploited the separable structure of blur kernels to initialize the network instances. Zhang et al. [37] integrated model optimization and deep learning via half quadratic splitting and then applied a CNN denoiser to tackle the problem of denoising. Zhang et al. [38] trained a fully CNN in the gradient domain to learn the image prior and employed network cascading with a deconvolution module to iteratively deblur images. In [37,38], the handcrafted prior was avoided, and a flexible approach to handling different vision problems was proposed. The authors addressed deblurring by learning a direct mapping from the degraded image to the ground truth, but the learned prior was used for denoising, whereas the blur kernel went almost unused in the network.

In general, deep learning retrieves a mapping function obtained from supervised learning and requires paired data (e.g., blurred image and its ground truth) for training. However, the ground truth is often unavailable in practice, thus limiting its applicability to blind natural image deblurring [39]. Overall, optimization and deep learning can be seen as complementary approaches. In fact, model optimization can be applied to a variety of vision tasks without requiring ground truth, whereas deep learning omits handcrafted priors and provides fast testing speed for deblurring images [40].

## 2.3. Outlier Handling

The complicated digital imaging pipeline influences the final visual quality, which is subject to external and internal factors, including illumination, exposure, camera lens, and sensors. Therefore, outliers produced by saturated pixels, non-Gaussian noise, camera nonlinear response, among others, tend to further degrade a blurred image, producing considerable ringing artifacts during deconvolution [41]. For instance, the sheet metal of an object under inspection can be very reflective and saturate pixels. Thus, suitable image deblurring is essential for dynamic visual inspection of these types of objects.

Whyte et al. [42] embedded a nonlinear degradation model into the RL deconvolution to discard saturated pixels by assigning low weights to them. Cho et al. [9] computed the mask specifying inliers and outliers during expectation and restored the image using hyper-Laplacian prior during maximization in the expectation–maximization algorithm. Hu et al. [43] incorporated the mask proposed in [9] to modify the RL deconvolution and combined the approaches in [42] and [9]. Moreover, the approaches in [35,36] added three convolutional layers at the end of their respective networks to reject outliers.

For dynamic visual inspection, we propose a novel method for blind image deblurring comprising blur kernel estimation and deconvolution modeling. Features in the transform domain were first selected to effectively estimate the blur kernel parameters assuming linear motion, thus avoiding rotation and error propagation. Then, the CNN-learned prior was embedded into the conventional objective function, where deconvolution was split into deblurring and denoising stages. Saturated pixels from metallic reflection were handled during deblurring. Instead of a handcrafted prior, we used a prior learned via CNN for denoising at a wide range of noise levels. Experimental results demonstrate that the proposed method can improve dynamic visual inspection and retrieves higher SSIM and FSIM than existing approaches.

## 3. Overview of Deblurring for Dynamic Visual Inspection

### 3.1. Dynamic Visual Inspection System

Figure 1 illustrates the dynamic visual inspection system considered in this study. Four LED strips are located at the sides of the object under inspection and intersect with the axis of the imaging plane. The object under inspection is surrounded by three cameras installed at the sides and on top of the assembly line and pointing to the object. The camera lens and light sources are intended to guarantee high visual quality. In the imaging pipeline for dynamic visual inspection, the object moves with respect to the cameras, leading to motion blur, especially appearing as smears on sharp edges. In turn, blurred images may lead to unreliable inspection results. Thus, we aimed to develop blind image deblurring for dynamic visual inspection to restore a suitable latent image and mitigate motion blur.

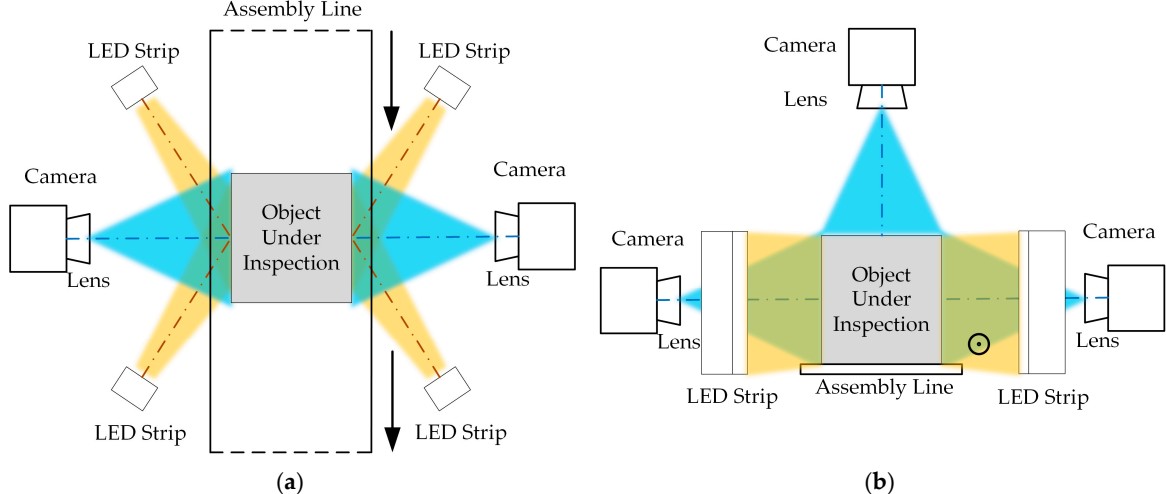

**Figure 1.** Diagram of dynamic visual inspection system: (**a**) top and (**b**) front view.

### 3.2. Blur Mitigation

Based on a spatially invariant blur kernel, we formulated the blur model as the latent image convolved with the blur kernel plus additive white Gaussian noise. As convolution is a linear operator, we express blurring as follows:

$$b = \mathbf{H}f + \eta, \tag{1}$$

where $b$, $f$, and $\eta$ are vectors representing the blurred image, latent image, and noise, respectively, and matrix $\mathbf{H}$ represents the blur kernel.

Blind image deblurring restores the latent image from its blurred version provided an accurate blur kernel estimate is available. The distance between the latent and restored images reflects the deblurring performance.

During the exposure time, the object under inspection moves along the assembly line, resulting in linear motion blur, which can be modeled as follows:

$$\mathbf{H}(x, y) = \begin{cases} \frac{1}{L} & \sqrt{x^2 + y^2} \leq \frac{L}{2}, \frac{y}{x} = \tan \theta \\ 0 & \text{otherwise} \end{cases}, \tag{2}$$

where $x$ and $y$ denote the pixel indices in the horizontal and vertical directions, respectively. Hence, the linear-motion blur kernel is described by two parameters, namely, blur angle $\theta$ and blur length $L$. The blur angle indicates the motion direction of the object with respect to the camera, and the blur length characterizes the distance that the object moves over the camera in pixels (px).

To guarantee high image quality for dynamic visual inspection, we estimated the blur kernel in the transform domain of the blurred image and then deblurred the image considering the presence of outliers based on the estimated kernel.

## 4. Proposed Method

In this section, we detail our method of blind image deblurring for dynamic visual inspection, including blur kernel estimation and deconvolution modeling. The blur kernel is estimated via features selected from the transform domain, and the deconvolution iteratively performs deblurring and denoising, where the former aims to reject outliers, and the latter is based on a CNN.

### 4.1. Blur Kernel Estimation

In our system (Figure 1), the heading of the object with respect to the cameras is fixed, and hence the blur angle is assumed to be known in advance (either 0° or 90°). Thus, the blur length is the only parameter to estimate for obtaining the blur kernel. First, we preprocessed the blurred image to select sharp edges and then estimated the blur length based on feature points from the resulting autocorrelation function.

#### 4.1.1. Sharp Edge Selection

We performed preprocessing to enhance sharp edge extraction and the subsequent blur kernel estimation. To this end, we filtered the blurred image with the Sobel detector to efficiently determine weighted edges for mitigating smear:

$$\nabla b = b \otimes \begin{bmatrix} 1 & 2 & 1 \\ 0 & 0 & 0 \\ -1 & -2 & -1 \end{bmatrix}, \tag{3}$$

where $\nabla b$ is the preprocessed image, and $\otimes$ denotes the convolution operator. The Sobel detector and its transpose detect vertical and horizontal edges, respectively.

Figure 2 shows examples of preprocessing for dynamic visual inspection, where the sharp edges in the blurred images are enhanced. The selected sharp edge has a scale corresponding to the blur length we aimed to estimate.

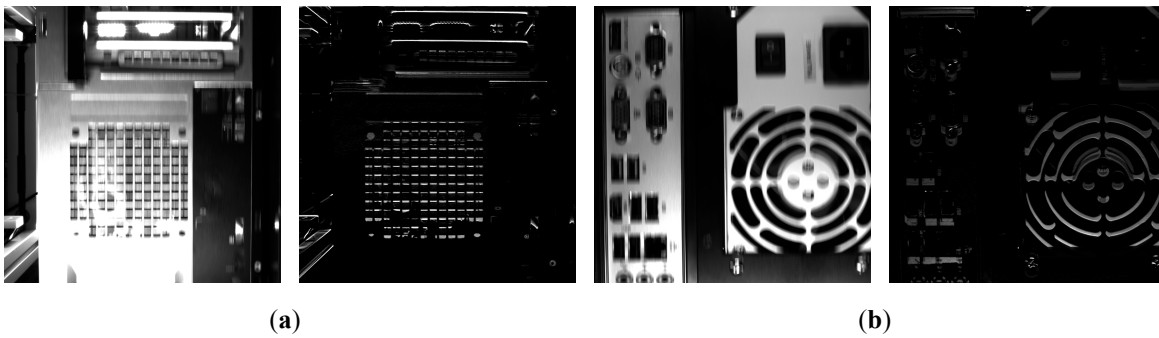

(**a**)                                                                                    (**b**)

**Figure 2.** Examples of preprocessing for dynamic visual inspection. The examples in (**a**) and (**b**) show the blurred image (left) and the result of preprocessing (right).

### 4.1.2. Blur Length Estimation

Once a sharp edge was selected, we estimated the blur length based on the autocorrelation function of the preprocessed blurred image as follows:

$$\Phi = \mathcal{FFT}^{-1}\big\{\big\|\mathcal{FFT}(\nabla b)\big\|\big\}, \tag{4}$$

where *FFT* and *FFT*$^{-1}$ denote the Fourier transform and its inverse, respectively, and $\Phi$ is the autocorrelation function with the same size as the blurred image. To visualize function $\Phi$, we plotted its columns according to the row in a plane, where the x-axis scale ranged from 1 to the number of rows in $\Phi$. This visualization results in conjugate pairs of valleys that retrieve sinc-shaped curves in the autocorrelation function, where the lowest pair corresponds to the blur length.

For instance, the images with sharp edges in Figure 2 retrieve the visualization of the autocorrelation function shown in Figure 3. A conjugate pair of valleys is clear in each plot, whereas other characteristics might be related to noise and outliers.

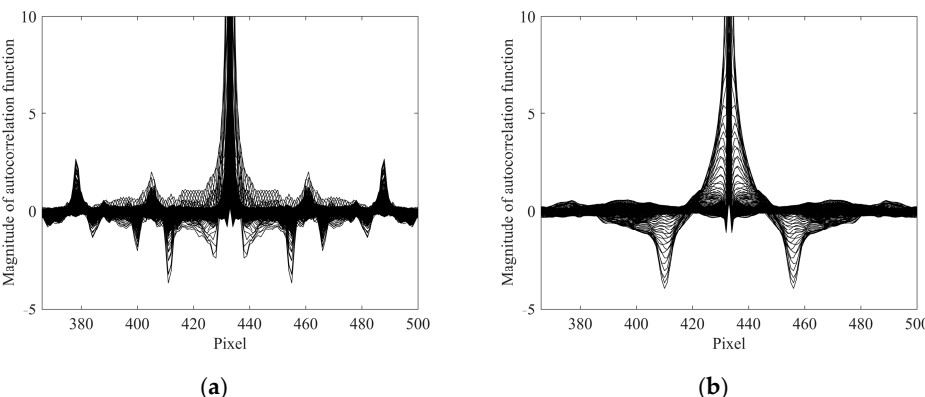

(**a**)                                                                 (**b**)

**Figure 3.** Autocorrelation function of preprocessed images in Figure 2. The functions plotted in (**a**) and (**b**) correspond to the respective images in Figure 2.

Given the known blur angle of 90° for dynamic visual inspection, blur length estimation is based on 1D data and does not need additional operations. Specifically, let $V_1(x_1, z_1)$ and $V_2(x_2, z_2)$ be the two lowest conjugate valleys, where $x$, $z$ respectively denote the horizontal pixel indices and the magnitude of its autocorrelation function. The half interval between these valleys corresponds to the blur length as follows:

$$L = \frac{|x_1 - x_2|}{2}. \tag{5}$$

*4.2. Deconvolution Modeling*

In principle, deconvolution can reverse blurred image degradation after blur kernel estimation through non-blind deblurring. Conventional optimization relies on image prior-based regularization, but either very strong or weak regularization can undermine the deblurring performance. Furthermore, such optimization is often time consuming because of the sophisticated and non-convex regularization. In contrast, deep learning can overcome these problems as it does not require handcrafted priors and provides fast testing speed.

4.2.1. Deconvolution Model Splitting

Conventional optimization is formulated via an objective function that includes data fidelity and regularization terms. The data fidelity term is modeled based on the noise distribution, and the regularization term constrains the solution space. From a Bayesian perspective, the best restored image is that optimizing the following objective function:

$$\hat{f} = \underset{f}{\text{argmin}} \frac{\lambda}{2} \|b - \mathbf{H}f\|_2^2 + \Theta(f), \tag{6}$$

where the first term is the data fidelity weighted by $\lambda$ and assuming Gaussian noise, and $\Theta(f)$ represents the image prior-based regularization.

Regularization is often non-convex and hence hinders optimization. To relax the model with non-convex regularization, we adopted variable splitting to decouple the objective function into individual subproblems. Among several variable splitting approaches, we used half-quadratic splitting [44] given its faster convergence than methods such as alternating direction method of multipliers (ADMM) [45] and primal-dual (PD) [46]. Hence, the deconvolution model is split as follows:

$$\underset{f,l}{\text{argmin}} \frac{\lambda}{2} \|b - \mathbf{H}f\|_2^2 + \Theta(l) + \beta\|f - l\|_2^2, \tag{7}$$

where $l$ is a slack variable approximately equal to $f$ and $\beta$ is a weight. Equation (7) can be alternately solved via the following iterative process:

$$\hat{l}_{t+1} = \underset{l}{\text{argmin}} \beta\|f_t - l\|_2^2 + \Theta(l), \tag{8}$$

$$\hat{f}_{t+1} = \underset{f}{\text{argmin}} \frac{\lambda}{2} \|b - \mathbf{H}f\|_2^2 + \beta\|f - l_t\|_2^2. \tag{9}$$

Equation (8) represents a deconvolution model, whose degradation is the identity matrix, and corresponds to Gaussian denoising. In addition, Equation (9) is convex and can be efficiently solved in the frequency domain of the Fourier transform. Therefore, half-quadratic splitting turns the complicated and non-convex deconvolution model into easier problems of denoising and deblurring given by Equations (8) and (9), respectively.

4.2.2. Outlier Handling

Deblurring in Equation (9) can be efficiently solved due to the convex $L_2$ norm, but outliers induced by the digital imaging pipeline can affect the goodness of fit in the data fidelity term and undermine the deblurring performance. Moreover, the sheet metal of the object under inspection considered in this study can intensely reflect light, thus saturating pixels in the images. Figure 4 shows a restoration example using the RL deconvolution [22] for dynamic visual inspection. The restored image contains severe ringing artifacts, especially at its salient edges, because saturated pixels behave as outliers, undermining the performance of approaches based on linear degradation [41].

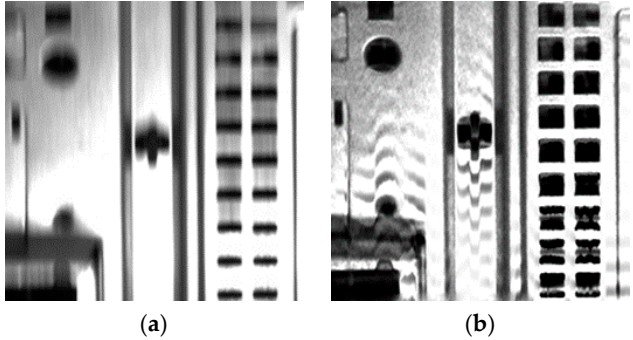

(a)          (b)

**Figure 4.** Deblurring with resulting ringing artifacts. (**a**) Blurred image from dynamic visual inspection system and (**b**) its restoration via the Richardson–Lucy (RL) deconvolution.

To handle saturated pixels, we first classified image intensities into saturated and unsaturated by using the binary mask in [9], where saturated pixels were rejected via low weights during deconvolution. To compute its expectation, the likelihood term associated with mask $m$ is formulated as follows:

$$p(b|m, \mathbf{H}, l) = \begin{cases} \mathcal{N}(b|\mathbf{H}l, \sigma) & \text{if } m = 1 \\ C & \text{if } m = 0 \end{cases}, \tag{10}$$

where $\mathcal{N}$ is a Gaussian distribution with standard deviation $\sigma$ and $C$ is the constant defined as the inverse of the dynamic range. Then, weight $W_t$ in the mask is given by the following:

$$W_t = \frac{E[m_t]}{2\sigma^2}, \tag{11}$$

$$E[m_t] = \begin{cases} \frac{\mathcal{N}(b|\mathbf{H}l_t, \sigma)p_{\mathrm{u}}}{\mathcal{N}(b|\mathbf{H}l_t, \sigma)p_{\mathrm{u}} + Cp_{\mathrm{s}}} & \text{if } \mathbf{H}l_t \in \mathrm{DR} \\ 0 & \text{otherwise} \end{cases}, \tag{12}$$

where $p_u$ and $p_s$ are the probabilities of a pixel being unsaturated and saturated, respectively, and $t$ is the iteration indices.

Finally, by letting $\gamma = \lambda/2\beta$ and including $W_t$ into Equation (9), we obtain the following:

$$\begin{aligned} \hat{f}_{t+1} &= \underset{f}{\mathrm{argmin}} \frac{\lambda W_t}{2} \|b - \mathbf{H}f\|_2^2 + \beta \|f - l_t\|_2^2 \\ &= \left(\gamma W_t \mathbf{H}^T \mathbf{H} + \mathbf{I}\right)^{-1} \left(\gamma W_t \mathbf{H}^T b + l_t\right) \end{aligned}. \tag{13}$$

Hence, $E[m_t]$ predicts the mask specifying unsaturated and saturated pixels in the Bayesian setting. Saturated pixels are not apt for the linear degradation model, thereby affecting the goodness of fit in the data fidelity term. By associating the data fidelity term to the mask, saturated pixels are rejected during deconvolution, and outliers can be handled during deblurring.

### 4.2.3. FFDNet Denoising

As Equation (8) corresponds to Gaussian denoising, we can rewrite it as follows:

$$\hat{l}_{t+1} = \underset{l}{\mathrm{argmin}} \frac{1}{2} \left( \sqrt{\frac{1}{2\beta}} \right)^{-2} \|f_t - l\|_2^2 + \Theta(l). \tag{14}$$

Hence, denoising processes image $f_t$ with Gaussian noise level $\sqrt{1/2\beta}$, and $\Theta(l)$ denotes the image prior. There are two kinds of image priors that are widely used in deconvolution—one based on local smoothness and the other characterizing nonlocal self-similarity. These priors can retrieve

suitable results, but they might not be strong enough for complex structures and textures. Furthermore, regardless of the time-consuming optimization, these priors must deal with excessive smoothing and block-like artifacts.

Recently, CNNs have shown promising results to efficiently learn image priors given the parallel computation ability of graphics processing units, and refined techniques to train and design CNNs facilitate their application for image restoration [38]. Thus, we use a CNN-based prior for denoising. Many CNN priors are trained with paired data at a fixed noise level, thus lacking flexibility. For dynamic visual inspection, the CNN prior should be determined at different noise levels and even handle the possibility of spatially variant noise. FFDNet can model a noise-level map as input to match varying noise levels, thus generalizing single CNN priors [10]. Table 1 lists the settings of FFDNet used in this study. The CNNs of grayscale and color images have 15 and 12 convolutional layers, respectively. Each layer basically consists of convolutions (Conv), rectified linear units (ReLU), and batch normalization (BN). The grayscale and color images employ 64 and 96 convolution kernels, respectively, and zero padding maintains fixed the size of feature maps.

**Table 1.** Fast and flexible denoising network (FFDNet) settings used in this study.

| Grayscale Image | Color Image |
|:---:|:---:|
| 15 convolutional layers | 12 convolutional layers |
| Downsampling + noise-level map ||
| (Conv 3 × 3 + ReLU) × 1 layer | (Conv 3 × 3 + ReLU) × 1 layer |
| (Conv 3 × 3 + BN + ReLU) × 13 layers | (Conv 3 × 3 + BN + ReLU) × 10 layers |
| (Conv 3 × 3) × 1 layer | (Conv 3 × 3) × 1 layer |
| Upscaling ||

Figure 5 illustrates the proposed blind deblurring of saturated images for dynamic visual inspection. After blur kernel estimation, a predicted mask was employed during deblurring to discard saturated pixels and prevent ringing artifacts. Instead of specifying a handcrafted prior, we trained the FFDNet at a wide range of noise levels, and the image prior was parameterized as weights in the network for denoising. Alternate deblurring and denoising were iteratively applied to fully deblur the image.

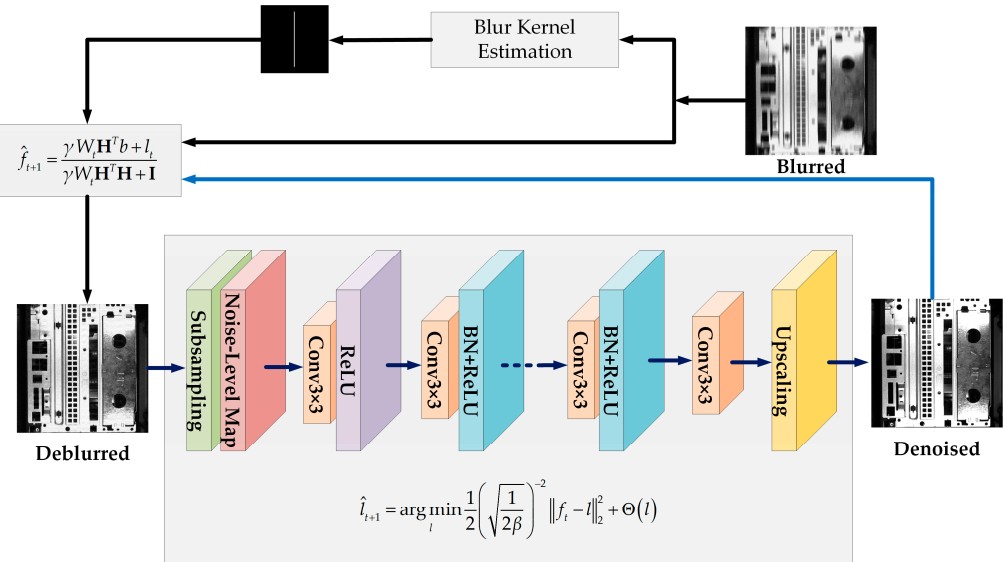

**Figure 5.** Diagram of proposed blind deblurring of saturated images for dynamic visual inspection. Deblurring handles saturated pixels, and denoising proceeds using image prior obtained from FFDNet. Conv, convolution; ReLU, rectified linear unit; BN, batch normalization.

Note that the proposed method can be extended to different vision tasks by setting matrix **H**. For instance, besides deblurring or denoising by setting **H** as the blur kernel or identity matrix, respectively, super-resolution can be achieved for **H** being a subsampling matrix [47].

## 5. Experimental Evaluation

We conducted experiments to validate the proposed blind deblurring method using the machine vision-based assembly line inspection system from Lite-On (Taipei, Taiwan) considering the objects under inspection to be computer cases Lenovo T280 G3, ADLINK EOS-1300, and Neousys NUVO-2400. We evaluated the proposed method through experiments on blur kernel estimation and deconvolution modeling.

### 5.1. Parameter Settings

We set standard deviation $\sigma$ in Equations (10)–(12) to 5/255 and $p_u$, $p_s$ to 0.9 and 0.1, respectively. Only $\lambda$ and $\beta$ were the unspecified parameters in Equation (13). Note that $\lambda$ depends on the noise level of the blurred image and remains fixed during deconvolution, whereas $\beta$ regularizes the FFDNet-learned prior, which can be implicitly determined by its noise level. The noise level of the prior, $\sqrt{1/2\beta}$, exponentially decays from 25/255 to the interval [1/255, 10/255] according to the number of iterations set to 30. Based on the original training set of FFDNet, we randomly selected 10% of the samples and increased pixel intensities by 30% to simulate saturation. To prevent adverse effects on the $L_2$ norm by outliers, we modified the loss function of FFDNet using the $L_1$ norm as follows:

$$\mathcal{L}(\varphi) = \frac{1}{N}\sum_{i=1}^{N} \|\mathcal{F}(x_i, \boldsymbol{M}_i; \varphi) - x_i^0\|_1^1,$$

(15)

where $\varphi$ represents the FFDNet weights, $\mathcal{F}$ is the mapping function, and $x$ and $x^0$ denote the degraded image and its ground truth, respectively, $N$ is the number of samples used for training per batch and $\boldsymbol{M}$ denotes the noise-level map of the FFDNet.

### 5.2. Blur Kernel Estimation

As the ground truth of the blur kernel remains unknown during dynamic visual inspection, we adopted the classical RL deconvolution [22] to deblur images based on the estimated blur kernel and validate the estimation detailed in Section 4.1. Figure 6 show examples of blurred computer case images obtained from dynamic visual inspection and their restorations. Figure 7 shows magnified views of these images. The estimated blur lengths are 28, 20, 22, and 27 px at fixed blur angle (90°) for the images in Figure 6, respectively. Excluding the ringing artifacts around the saturated pixels, motion blur is mostly removed, thus visually validating the proposed blur kernel estimation.

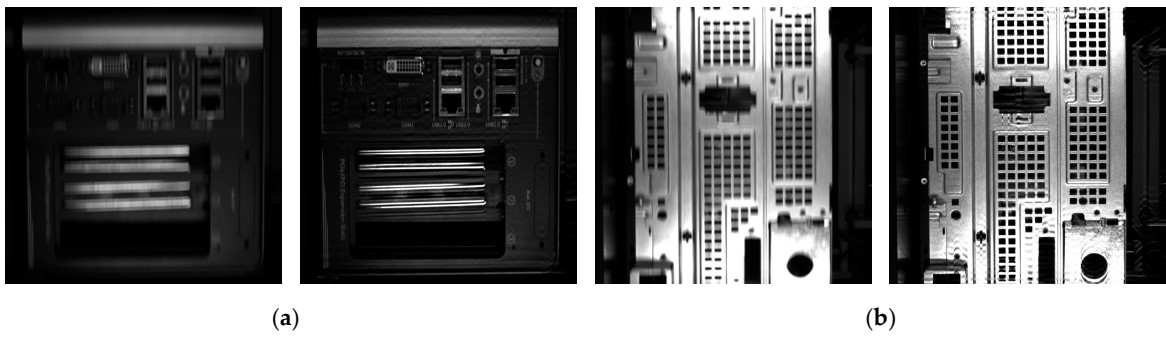

(**a**)      (**b**)

**Figure 6.** *Cont.*

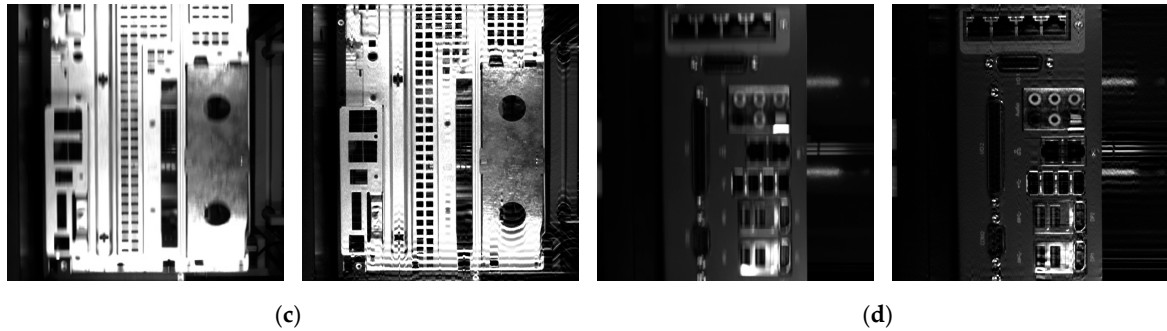

(**c**)            (**d**)

**Figure 6.** Blind image deblurring for dynamic visual inspection based on estimated blur kernel. The estimated blur lengths are (**a**) 28 px, (**b**) 20 px, (**c**) 22 px, and (**d**) 27 px. The blurred images and their restored versions are shown side by side.

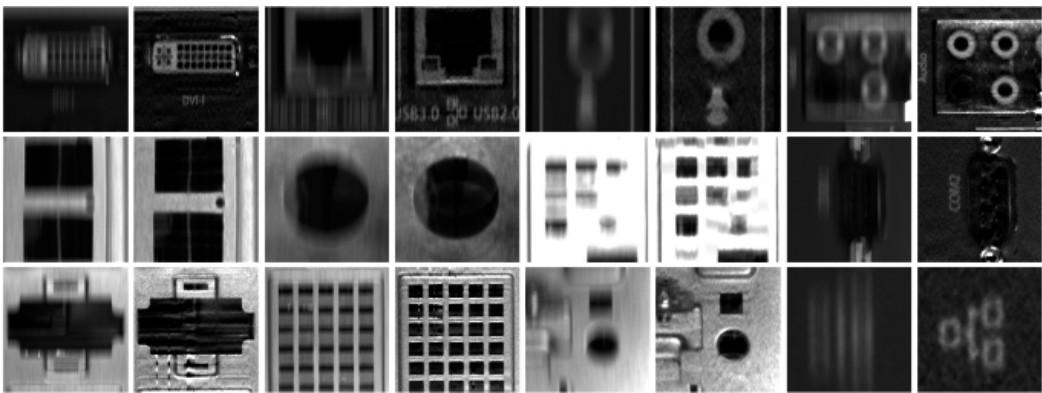

**Figure 7.** Magnified views of examples in Figure 6. The blurred images and their restored versions are shown side by side.

*5.3. Deconvolution Modeling*

Although the proposed blur kernel estimation improves image quality, ringing artifacts attributed to saturated pixels persist. Hence, we further validated the proposed deconvolution model and compared it to existing approaches, namely, RL [22], IDD-BM3D [28], NCSR [29], HL [26], fully convolutional neural network (FCNN) [37], image restoration convolutional neural network (IRCNN) [38], and VEM [9], for dynamic visual inspection. For a fair comparison, we fine-tuned the parameters from those approaches and used the pretrained models for FCNN and IRCNN.

Figure 8 visually compares the restored images using the proposed and similar methods. Regarding visual quality, IDD-BM3D and NCSR slightly outperform the RL deconvolution, but those two methods collect similar image blocks nonlocally, and small structures, especially text, are not suitably restored where block-like artifacts occur. HL uses a hyper-Laplacian prior to model a heavy-tailed distribution, but its performance reduces in the presence of saturated pixels, leading to severe ringing artifacts. FCNN and IRCNN are deep learning methods that fail to effectively handle outliers. Moreover, FCNN needs retraining according to the noise level, and residual blur remains over the text. VEM is tailored for handling saturated pixels, thus providing a highly improved visual quality in spite of a few visible particles. Compared with VEM, the proposed method with CNN-based prior further denoises and smooths the restored image, thus reducing noise and highlighting fine details.

Measures such as the peak signal-to-noise ratio and mean squared error cannot be applied to dynamic visual inspection, as no ground truth is available as reference. On the other hand, measures without reference are not suited for comparison with human perception [48]. Thus, we combined reference-based measures with subjective visual quality evaluations to determine the deblurring performance. Specifically, we first excluded the results whose subjective visual quality was obviously

low, and for the rest of the results, we applied blurring to the restored images, that is, we subjected the restored image $\hat{f}$ to the estimated blur kernel $\hat{\mathbf{H}}$ to synthesize blurred image $\hat{b}$ ($\hat{b} = \hat{\mathbf{H}}\hat{f}$) from the original blurred image $b$. Then, we adopted two reference-based measures, namely, SSIM [49] and FSIM [50], to determine the difference between $\hat{b}$ and $b$ for obtaining the deblurring performance, while the conventional measures were based on the restored image $\hat{f}$ and its ground truth $f$, where $f$ is unavailable in our case.

Table 2 lists the quantitative results from comparing restorations using RL, IRCNN, VEM, and the proposed method. The results from IDDBM3D, NCSR, HL, and FCNN are excluded from the table given their low subjective visual quality. It can be seen that the proposed method outperforms the existing approaches, especially for the images in Figure 8b,c which contain severe pixel saturation. The increase in scores ranges from 0.09% (FSIM of restored image using our method versus that using VEM in Figure 8b) to 19.62% (SSIM of restored image using our method versus that using RL in Figure 8c).

For simplicity, the blur kernel is assumed to be spatially invariant. However, the blur kernel is in fact spatially variant, because the background remains static during exposure time. Including the proposed method, the restoration in Figure 8d using the evaluated deblurring methods contains severe ringing artifacts over the background.

As the RL deconvolution does not involve regularization, it might not suitably reflect the inherent nature of the image for dynamic visual inspection. Consequently, less severe ringing artifacts occur over the background in its restoration, and it scores higher on the employed measures at the expense of large noise.

Table 3 reports the average run times (in seconds) on the restorations of Figure 8 (856 × 856). RL, IRCNN, VEM, and the proposed method were run on a Nvidia GeForce GTX 1080 (Santa Clara, CA, USA). One can see that the proposed method avoids the time-consuming optimization by incorporating the FFDNet-learned prior, thus delivering a faster speed than VEM. Compared with IRCNN, the proposed method sacrifices a little speed for a better deblurring performance. The simple and efficient RL deconvolution achieves a very competitive speed, but it does not make a good compromise between speed and performance, as is shown in Figure 8 and Table 2.

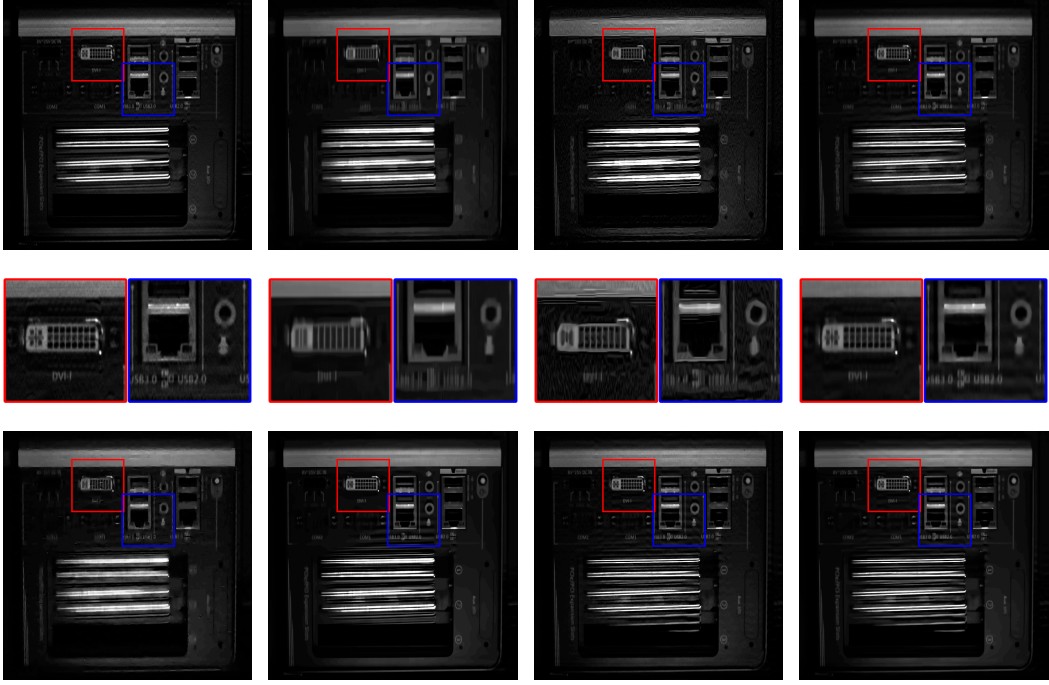

**Figure 8.** *Cont.*

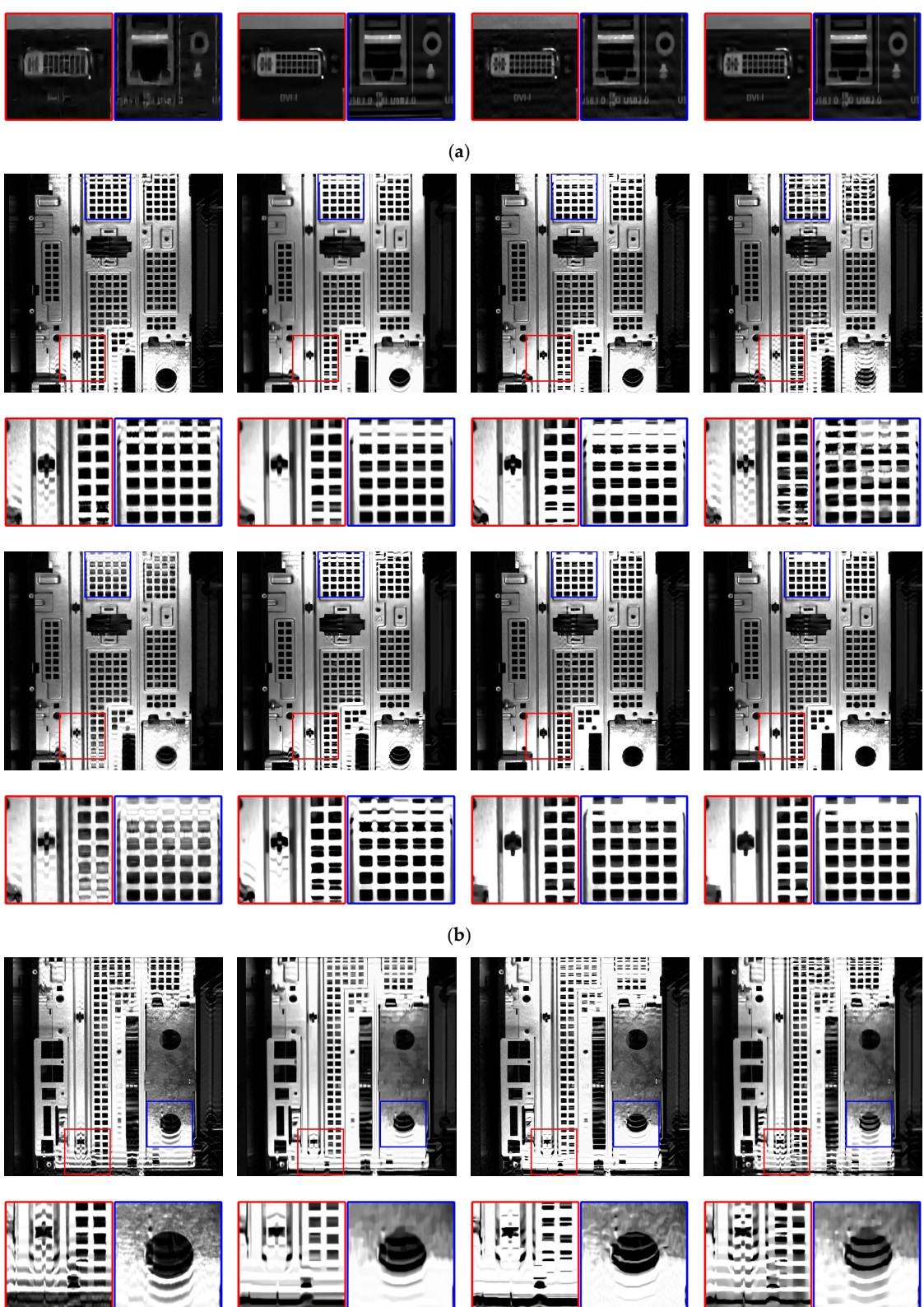

**Figure 8.** *Cont.*

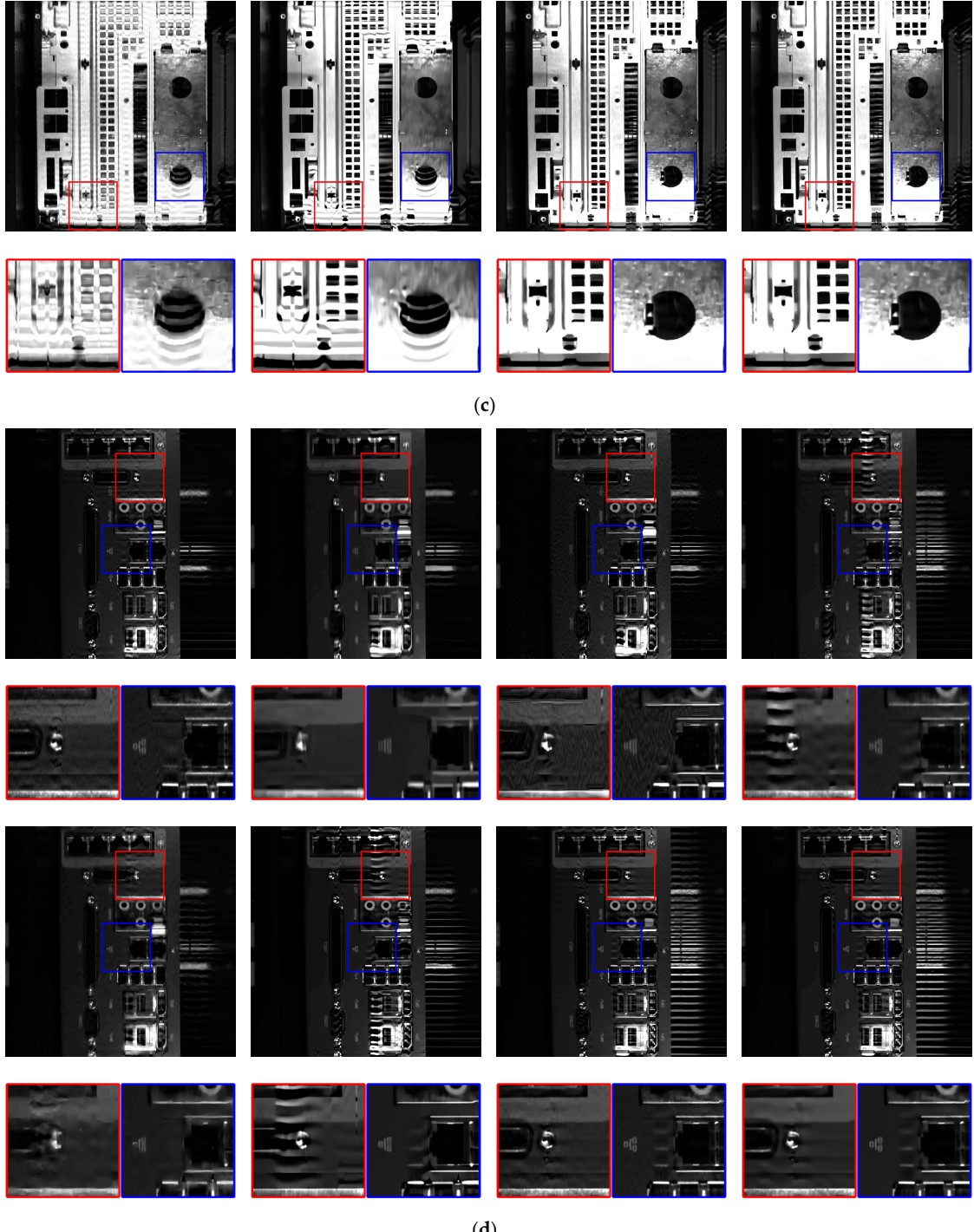

(**c**)

(**d**)

**Figure 8.** Restorations including magnified views of the blurred images in Figure 6 for dynamic visual inspection using different methods where the estimated blur lengths are (**a**) 28 px, (**b**) 20 px, (**c**) 22 px, (**d**) 27 px. In the left to right and top-down order, deblurring was performed by applying RL [22], iterative decoupled deblurring-block matching and 3D filtering (IDD-BM3D) [28], nonlocally centralized sparse representation (NCSR) [29], hyper-Laplacian (HL) [26], fully convolutional neural network (FCNN) [37], image restoration convolutional neural network (IRCNN) [38], variational expectation–maximization (VEM) [9], and our proposed method.

**Table 2.** Score comparisons of the structural similarity index (SSIM) and feature similarity index (FSIM) on the restorations of Figure 8.

| Figure 8 | (a) | (b) | (c) | (d) | (a) | (b) | (c) | (d) |
|---|---|---|---|---|---|---|---|---|
| Measures | SSIM | | | | FSIM | | | |
| RL [22] | 0.6932 | 0.6467 | 0.6126 | 0.7337 | 0.7586 | 0.7675 | 0.7384 | 0.8140 |
| IRCNN [38] | 0.7208 | 0.7178 | 0.7300 | 0.7030 | 0.7587 | 0.7685 | 0.7685 | 0.7390 |
| VEM [9] | 0.7094 | 0.7279 | 0.7198 | 0.6911 | 0.7384 | 0.7970 | 0.7864 | 0.7063 |
| Our method | 0.7263 | 0.7417 | 0.7328 | 0.7055 | 0.7407 | 0.7977 | 0.7889 | 0.7093 |

**Table 3.** Average run time (in seconds) on the restorations of Figure 8.

| Methods | RL [22] | IRCNN [38] | VEM [9] | Our Method |
|---|---|---|---|---|
| Figure 8 (856 × 856) | 2.03 | 6.83 | 18.95 | 7.26 |

## 6. Conclusions

We propose a method to perform blind deblurring of saturated images for dynamic visual inspection. Given the assembly line application, in the proposed method, we considered the blur angle of the linear motion as known and fixed, and only estimated the blur length from the autocorrelation function of selected sharp edges. Then, we employed variable splitting and included an FFDNet-learned prior into the objective function for optimization. Furthermore, saturated pixels were rejected via a binary mask associated with the data fidelity term, whereas a noise-level map ensured that the prior was applicable under a wide range of noise levels. The proposed method omitted saturated pixels and learned the image prior adaptively, effectively integrating optimization and deep learning. For deblurring performance evaluation, instead of using an original clear image as ground truth, we synthesized a blurred image from the restored one. Then, we determined the SSIM and FSIM, which indicated that the proposed method in general outperforms existing approaches, with improvements in the measures of 0.09–19.62%. As image degradation during dynamic visual inspection on an assembly line corresponds to linear motion with a static background, the blur kernel is spatially variant. Hence, in future work, we will focus on spatially variant blur kernel estimation to further improve the deblurring performance. In addition, as data fidelity is decoupled to handle saturated pixels, the fast testing speed of deep learning is not fully exploited, and further developments could improve the proposed method to deblur images in real time.

**Author Contributions:** B.W. and G.L. conceived the study. B.W. performed the experiments and wrote the paper with J.W. who also analyzed the results.

**Funding:** This research was funded by the Guangzhou Science and Technology Plan Project (201802030006) and the Open Project Program of Guangdong Key Laboratory of Modern Geometry and Mechanics Metrology Technology (SCMKF201801).

**Acknowledgments:** The authors would like to thank Lenovo and Dell, Inc. for providing several kinds of computer cases. They also thank Lite-On for building up and allowing them to use the visual inspection equipment on its computer case assembly line. The authors also thank Editage (www.editage.cn) for English language editing.

**Conflicts of Interest:** The authors declare no conflict of interest.

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
