# Peer review of "Blind Deblurring of Saturated Images Based on Optimization and Deep Learning for Dynamic Visual Inspection on the Assembly Line"

_symmetry, doi:10.3390/sym11050678_

Round 1

Reviewer 1 Report

The paper is well-written and nicely structured. I have only few minor remarks:

In equations (7) and (8) the arguments of the minimization are mistaken. For (7) it should be f, instead of l, while for (8) it should be both f,l, instead of only l.

On line 267, E[mt] should be changed to E[m_t] ("t" should be a sub-index)

There is too much repetition in the captions of the figures that can be avoided. For example, why don't the authors unite Figures 6-9 into one figure and only specify the value L for each of the images?

For more fair comparison analysis, maybe the authors can include the execution times of the four algorithms in Table 2 (or in a separate table)

Author Response

Point 1: In equations (7) and (8) the arguments of the minimization are mistaken. For (7) it should be f, instead of l, while for (8) it should be both f, l, instead of only l.. 

Response 1: Those mistakes are properly corrected.

Point 2: On line 267, E[mt] should be changed to E[m_t] ("t" should be a sub-index).

Response 2: This mistake is properly corrected, and we have checked all the symbols here and retyped them using mathtype.

Point 3: There is too much repetition in the captions of the figures that can be avoided. For example, why don't the authors unite Figures 6-9 into one figure and only specify the value L for each of the images?

Response 3: We united Figures 6-9 and Figures 11-14 into one figure respectively with L specified.

Point 4: For more fair comparison analysis, maybe the authors can include the execution times of the four algorithms in Table 2 (or in a separate table).

Response 4: We added the Table 3 containing the execution times of the four algorithms and analysed them in the text.

Reviewer 2 Report

1. As it is not known which score you mean in the abstract. I suggest to remove the numerical value.
2. The mask of Sobel operator is not an equation.
3. Improve the typesetting of (15): 1/x^2 -> x^(-2)
4. Modify Tab. 1 so that it does not contain the same rows or remove it and describe the structure in the text.
5. The test images 11-14 (6-9 ?) should be presented in one figure.
6. The experiments performed to measure the SSIM and FSIM measures should be better explained.

Author Response

Point 1: As it is not known which score you mean in the abstract. I suggest to remove the numerical value. 

Response 1: The numerical value of score improvements is removed.

Point 2: The mask of Sobel operator is not an equation.

Response 2: We rewrite Eqn. (3) specifying the Sobel operator and remove Eqn. (4).

Point 3: Improve the typesetting of (15): 1/x^2 -> x^(-2).

Response 3: The typesetting of (15) is improved, so is the Fig5 containing Eqn. (15).

Point 4: Modify Tab. 1 so that it does not contain the same rows or remove it and describe the structure in the text.

Response 4: Tab. 1 is modified to be more concise.

Point 5: The test images 11-14 (6-9 ?) should be presented in one figure.

Response 5: The test images 11-14 and 6-9 are now combined into one figure respectively.

Point 6: The experiments performed to measure the SSIM and FSIM measures should be better explained.

Response 6: The experiments performed to measure the SSIM and FSIM are modified to be more detailed.
